# Mesophyll porosity is modulated by the presence of functional stomata

Marjorie R. Lundgren [1,4], Andrew Mathers[2], Alice L. Baillie [1], Jessica Dunn [3], Matthew J. Wilson[1], Lee Hunt [3], Radoslaw Pajor[2], Marc Fradera-Soler [2], Stephen Rolfe [1], Colin P. Osborne [1], Craig J. Sturrock [2], Julie E. Gray [3], Sacha J. Mooney [2] & Andrew J. Fleming [1]

The formation of stomata and leaf mesophyll airspace must be coordinated to establish an efficient and robust network that facilitates gas exchange for photosynthesis, however the mechanism by which this coordinated development occurs remains unclear. Here, we combine microCT and gas exchange analyses with measures of stomatal size and patterning in a range of wild, domesticated and transgenic lines of wheat and Arabidopsis to show that mesophyll airspace formation is linked to stomatal function in both monocots and eudicots. Our results support the hypothesis that gas flux via stomatal pores influences the degree and spatial patterning of mesophyll airspace formation, and indicate that this relationship has been selected for during the evolution of modern wheat. We propose that the coordination of stomata and mesophyll airspace pattern underpins water use efficiency in crops, providing a target for future improvement.

[1] Department of Animal and Plant Sciences, University of Sheffield, Western Bank, Sheffield S10 2TN, UK. [2] Division of Agriculture and Environmental Sciences, School of Biosciences, University of Nottingham, Sutton Bonington Campus, Loughborough LE12 5RD, UK. [3] Department of Molecular Biology and Biotechnology, University of Sheffield, Western Bank, Sheffield S10 2TN, UK. [4] Lancaster Environment Centre, University of Lancaster, Lancaster LA1 4YQ, UK. Correspondence and requests for materials should be addressed to A.J.F. (email: a.fleming@sheffield.ac.uk)

Leaf photosynthetic function depends on the flux of $CO_2$ from the atmosphere to the primary sites of carbon fixation in the chloroplasts of mesophyll cells. To accommodate this flow of $CO_2$ in, and water vapour out, leaf surfaces have controllable stomatal pores that are connected to mesophyll cells via an intricate network of air channels[1]. This linkage between stomata and the airspace network was observed in the early microscopy observations of plants[2] and is accepted as a fundamental aspect of leaf structure, yet the mechanism by which stomatal formation and mesophyll airspace differentiation are coordinated to form a functional network of airspace is unknown. Previous work confirmed an inherent link between the two developmental processes[3,4], with a paucity of mesophyll airspace not deterring stomatal development[5,6], leading to the idea that stomata are initiated prior to the airspace network. Recent research shed new light on the cause and effect of these developmental events by suggesting that it is the development of stomata that modulates mesophyll airspace differentiation[7]. However, it remains unknown whether it is simply the presence of guard cells or actually the exposure of mesophyll cells to fluxes of $CO_2$ and $H_2O$ gases permitted by the presence of functional stomata that triggers mesophyll cell separation to create extensive intercellular airspaces.

We first investigate the association of stomata and mesophyll airspace by focusing on a series of wild and domesticated wheat lines. As a monocot grass, wheat leaves have a distinct cellular architecture with limited differentiation between adaxial and abaxial mesophyll, potentially simplifying the identification of relationships between mesophyll structure and stomata relative to eudicots. Moreover, modern domesticated wheat is hexaploid and derived from a complex series of evolutionary events between diploid and tetraploid progenitors[8]. Since ploidy level correlates with stomatal size and density[9–11], we reason that comparison of these diverse wheat relatives facilitates the identification of any correlation between stomata traits and mesophyll structure. We asses mesophyll airspace network structure by employing a micro X-ray Computed Tomography (microCT) approach developed to quantify mesophyll porosity and exposed internal surface area across the intact leaf[12,13]. Increasing the surface area of mesophyll tissue is expected to facilitate high exchange rates of $CO_2$ and $H_2O$ between stomata and chloroplasts[13–17], and airspace connectivity is linked to effective gas diffusion within the leaf[18]. We couple these assessments of stomatal patterning and leaf cellular structure with measures of stomatal conductance to test the hypothesis that changes in mesophyll airspace are related not only to the presence but also the function of stomata in gas exchange. Since it has been proposed that distinct mechanisms underpin stomatal patterning in monocot and eudicot systems[19], we extend our analysis in wheat to the eudicot Arabidopsis, exploiting a range stomatal development mutants to identify a fundamental mechanism linking stomatal and mesophyll differentiation.

Our results indicate, firstly, that the degree and extent of separation of mesophyll cells to form airspaces is linked to the presence of functional pores (i.e., stomata that allow gas flux), rather than simply relying on the presence of differentiated guard cells. Secondly, they suggest a step-wise selection during wheat evolution for leaves with a decreased stomatal density/increased stomatal size that is associated with both a decrease in $g_s$ and a decrease in mesophyll porosity, yielding a denser leaf. We thus clarify the link between stomata and mesophyll airspace development in both monocot and eudicot leaf systems and provide new insights into the process of wheat leaf evolution.

## Results

**Wheat stomatal size and density changes with ploidy**. To investigate whether stomatal characteristics vary with ploidy level in wheat, we examined the size and patterning of stomata in two diploid (2n) species (*Triticum baeoticum* and *T. urartu*), two tetraploid (4n) species (*T. araraticum*, *T. dicoccoides*) and three cultivars of the hexaploid (6n) *T. aestivum* (cv. Cougar, Crusoe and Shango). The wheat species display characteristic grass stomata, with the stomatal complexes (each composed of a pair of guard cells flanked by subsidiary cells) lying in epidermal cell files along the leaf surface. Example images showing the overall distribution of stomata in the different ploidy backgrounds are shown in Fig. 1a–c, with higher resolution images of individual stomatal complexes shown in Fig. 1d–f. These images suggest that leaf stomatal size and density is influenced by ploidy in wheat. Measurement of these parameters followed by statistical analysis (ANOVA with posthoc Tukey) showed that stomatal complexes of the hexaploid cultivars were wider than those of both tetraploid ($P < 0.001$) and diploid species ($P < 0.001$), which were themselves indistinguishable ($P = 0.115$; Fig. 1g). In contrast, the length of stomatal complexes was indistinguishable between tetraploid and hexaploid lines (ANOVA with posthoc Tukey, $P = 0.479$), while stomata were significantly shorter in the diploid wheats than both tetraploid ($P < 0.001$) and hexaploid lines ($P < 0.001$; Fig. 1i). Thus, because stomatal complex area depends on both the length and width of stomata, the step-wise differences in these parameters between the three ploidy backgrounds leads to stomata size being distinct at each ploidy level (ANOVA with posthoc Tukey, diploid/tetraploid $P < 0.001$; tetraploid/hexaploid $P < 0.001$). Stomatal complexes were smallest in the diploid species, largest in the hexaploid cultivars, and intermediate in the tetraploid species (Fig. 1h). The step-wise increase in stomatal complex length shown in Fig. 1i was mirrored by stomatal density (Fig. 1j), with the diploid species having distinctly higher densities than observed in the tetraploid (ANOVA with posthoc Tukey, $P < 0.001$) and hexaploid species ($P < 0.001$) (which could not be distinguished from each other based on stomatal density; $P = 0.616$). Our data suggest that there has been an indirect selection for leaves with larger but relatively fewer stomata during the complex domestication of modern hexaploid wheat. This seems to have occurred in a step-wise fashion such that tetraploids are distinct from diploids in having longer, less dense stomata of similar width, and the hexaploid modern bread wheat cultivars having wider stomatal complexes than their tetraploid wild relatives, which likely occurred subsequent to or concurrent with the fusion of the diploid and tetraploid progenitors.

**Mesophyll airspace and stomata patterning are coordinated**. To investigate potential relationships between the observed variation in stomatal characteristics and mesophyll patterning, we used high-resolution microCT image analysis to quantify mesophyll porosity and surface area in the same wheat lines used for stomatal characterisation above. Example images of *T. urartu*, *T. araraticum* and *T. aestivum* cv Cougar are shown in Fig. 2 as 3D reconstructions of leaf segments (Fig. 2a–c), with exemplar transverse sections (Fig. 2d–f), longitudinal sections (Fig. 2g–i), and paradermal sections (Fig. 2j–l) in which airspace is represented in yellow and cellular material in green. Equivalent images for the other wheat lines analysed are shown in Supplementary Fig. 1. These views show that all of the wheat leaves display a classical grass leaf anatomy, consisting of parallel veins along the leaf longitudinal axis forming boundaries for mesophyll tissue. The cell separation within the mesophyll tissue forms a highly regular pattern of airspace that is clearly demonstrated in the longitudinal and paradermal views (Fig. 2g–l), while the sections shown in Fig. 2j–l provide an indication of differences in the size, distribution and overall amount of airspace between the wheat species. MicroCT provides a means to quantify these differences,

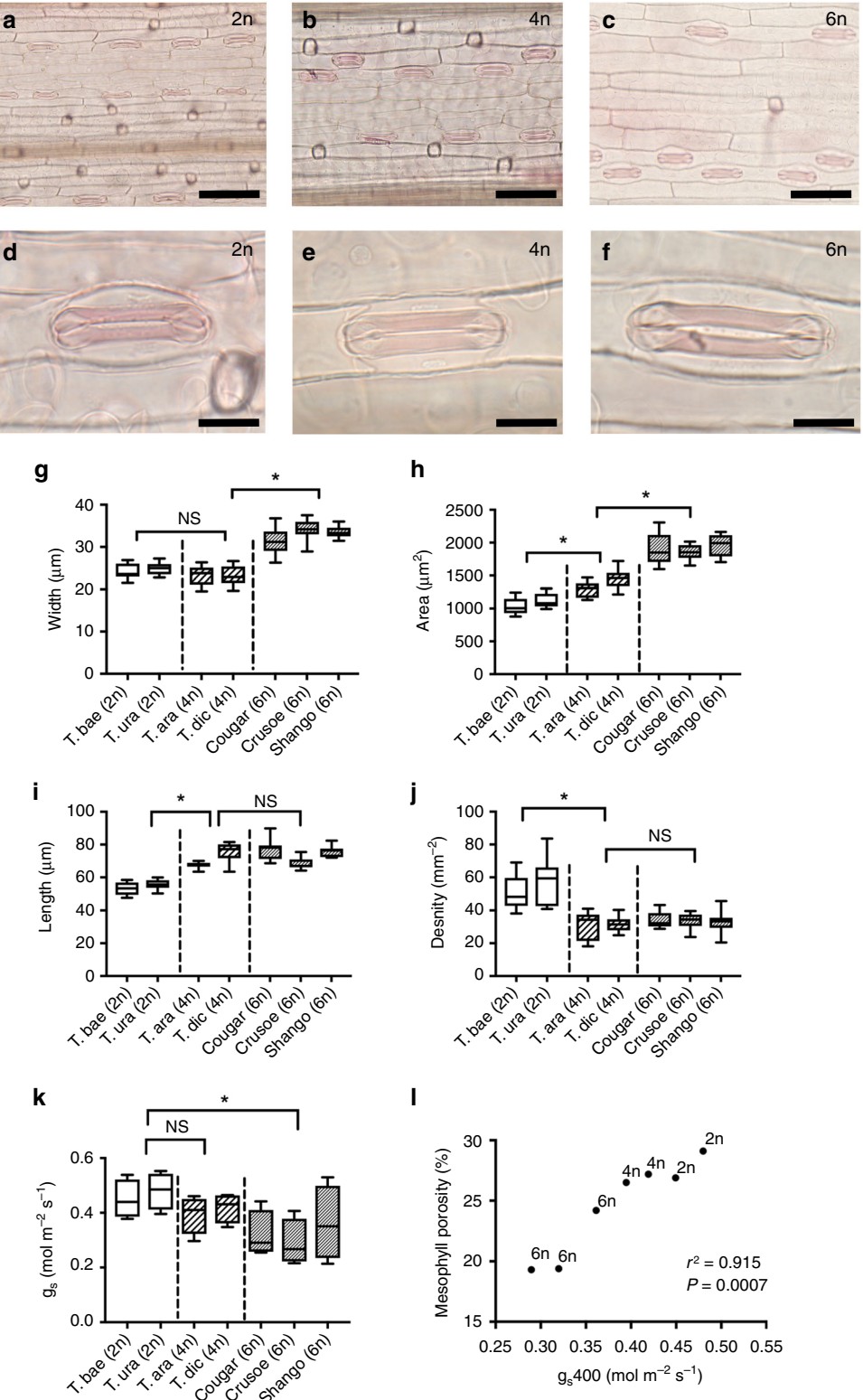

**Fig. 1** Stomatal patterning shifts with ploidy level in wheat. Sample images of overall stomatal distribution along the adaxial epidermis (**a–c**) and individual stomatal complexes (**d–f**) in *Triticum baeoticum* (2n; **a**, **d**), *T. araraticum* (4n; **b**, **e**) and *T. aestivum* cv Cougar (6n; **c**, **f**). Scale bar **a–c** = 80 μm; **d–f** = 20 μm. Stomatal width (**g**) (ANOVA, $F_{(2,81)} = 169.5$, $P < 0.0001$), area (**h**) (ANOVA, $F_{(2,81)} = 218.7$, $P < 0.0001$), length (**i**) (ANOVA, $F_{(2,73)} = 80.29$ $p < 0.0001$), density (**j**) (ANOVA, $F_{(2,81)} = 61.21$ $P < 0.0001$), and conductance, $g_s$ (**k**) (ANOVA, $F_{(2,25)} = 7.494$, $P = 0.0028$) are shown for all analysed wheat lines. Results of a posthoc Tukey test comparing sequential ploidy levels are indicated within each analysis, with an asterisk when significant at the $p < 0.05$ level or NS when not significant. For **g–k**, data are shown as box plots (25th−75th percentile, horizontal line = median) with whiskers indicating maximum and minimum values, $n = 6$. **l** For each analysed wheat line, mean mesophyll porosity is plotted against mean stomatal conductance, $g_s$, with ploidy level indicated for each point. Results of correlation analysis are presented (Pearson $r^2$ value). Results for individual paired data are shown in Supplementary Fig. 2

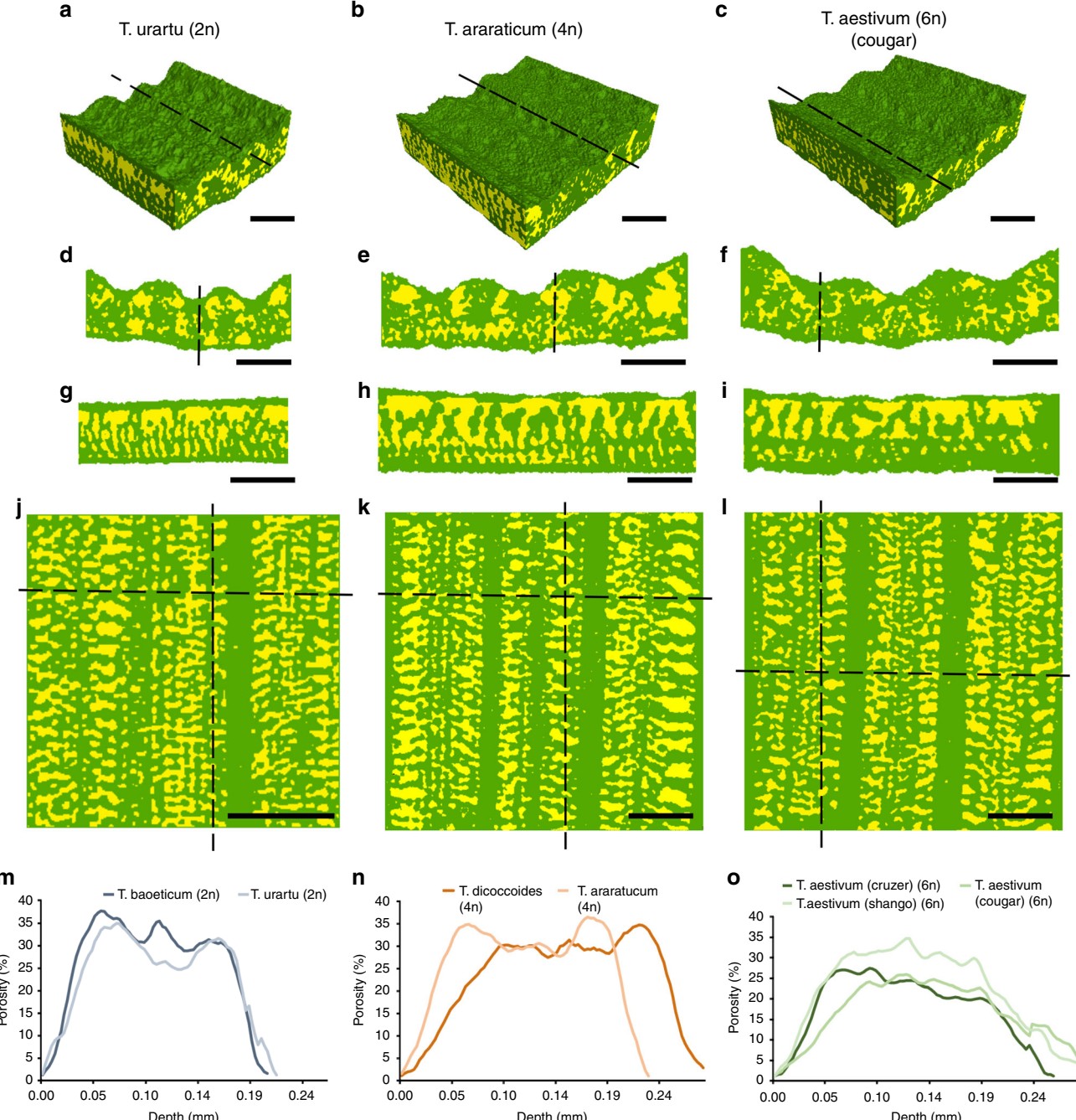

**Fig. 2** MicroCT imaging reveals variation in wheat leaf airspace with ploidy. Sample leaf images of *Triticum urartu* (2n), *T. araraticum* (4n) and *T. aestivum* cv Cougar (6n) in 3D renderings of tissue blocks (**a–c**), transverse sections (**d–f**), longitudinal sections (**g–i**), and paradermal sections (**j–l**), with solid tissue in green and airspace in yellow. Mesophyll porosity (%) (**m–o**) is plotted along leaf depth from adaxial to abaxial surfaces in the diploid (**m**), tetraploid (**n**), and hexaploid (**o**) lines, as indicated. *T. baoeticum*–dark blue; *T. urartu*–light blue; *T. dicoccoides*–dark orange; *T. araraticum*–light orange; *T. aestivum* (Crusoe)–dark green; *T. aestivum* (Cougar)–mid-green; *T. aestivum* (Shango)–light green. For clarity, only mean values of 6 replicated samples are presented in panels **m–o**. Lines in **a–c** indicate plane of section in **g–i**, respectively, also indicated by vertical lines in **j–l**. Horizontal lines in **j–l** indicate plane of section for **d–f**, respectively. Image resolution = 2.75 μm. Scale bars **a–l** = 200 μm

not simply in 2D sections but across the 3D depth of the tissue. Analysis of tissue porosity (i.e., airspace volume as a proportion of total tissue volume) from the adaxial (top) to the abaxial (bottom) surface of leaves revealed similarities and differences in the amount and distribution of airspace. All species displayed a pattern of increasing porosity further away from the epidermis, with a plateau of relatively high porosity across the middle part of the leaf (Fig. 2m–o). The rate of increase of porosity with distance into the leaf was greatest for the diploid species

(Fig. 2m) with the hexaploid species showing a shallower porosity gradient (Fig. 2o) and, generally, a lower maximal porosity value than that observed in the diploid species. The tetraploid species showed intermediate patterns of porosity throughout the leaf depth (Fig. 2n). Overall, our analysis of structural parameters across wheat leaves of different ploidy suggests that while the basic pattern of airspace and tissue distribution within the leaf has been conserved during the evolution of hexaploid wheat from diploid and tetraploid wild relatives, there has been

selection, whether direct or indirect, for a leaf structure with lower porosity (i.e., denser mesophyll).

**Gas exchange reflects airspace and stomatal patterning**. Using the diversity represented in these wheat relatives, we proceeded to investigate the effects of the observed trends in stomatal size/density and mesophyll airspace on gas exchange, via measurements of stomatal conductance to water vapour ($g_s$) and estimates of maximum stomatal conductance ($g_{smax}$). These data revealed a striking positive correlation between mesophyll porosity and $g_s$ ($r^2 = 0.915$, $P = 0.0007$; Fig. 1l) with diploid species showing high $g_s$ and high porosity and hexaploid leaves having a low $g_s$ and low porosity (Fig. 1k). This strong correlation of mesophyll porosity and $g_s$ was maintained when individual replicate plant paired data for the various lines analysed were considered (Pearson correlation, $r^2 = 0.451$, $P = 0.0001$; Supplementary Fig. 2). The decrease in porosity associated with increased ploidy level was also linked to a decrease in exposed mesophyll surface area per tissue volume, resulting in a strong positive correlation of $g_s$ and this parameter (Pearson correlation, $r^2 = 0.718$, $P = 0.016$; Supplementary Fig. 3a) which was also observed when exposed mesophyll surface area was expressed on a per leaf area basis (Pearson correlation, $r^2 = 0.633$, $P = 0.0323$; Supplementary Fig. 3b). The relationship of $g_{smax}$ (calculated using measurements as shown in Supplementary Fig. 3e) and porosity was less strong (Pearson correlation, $r^2 = 0.487$, $P = 0.081$; Supplementary Fig. 3c) than observed with $g_s$ (Fig. 1l). Analysis of $g_{smax}$ and stomatal area revealed an inverse correlation (Pearson correlation, $r^2 = 0.613$, $P = 0.037$) (Supplementary Fig. 3d) consistent with previous work indicating a complex trade-off between stomatal size and density on $g_s$, with leaves displaying high densities of smaller stomata conveying greater $g_s$ per stomatal area than leaves with low densities of larger stomata[20,21]. Overall, our analysis of $g_s$, mesophyll porosity and exposed surface area is consistent with the hypothesis that greater exposed mesophyll surface area and allocation of tissue volume to airspace facilitates increased gaseous diffusion to and from the mesophyll.

**The relationship between stomata and mesophyll airspace**. The analyses presented above are consistent with, but do not prove, a causal relationship between stomatal differentiation and mesophyll airspace formation. To test this hypothesis we used a series of transgenic lines of wheat with altered stomatal properties. Significant evidence shows that stomatal patterning is controlled via a series of mobile peptide signals, Epidermal Patterning Factors (EPFs)[22–24], which provide effective tools to alter stomatal density and, thus, investigate the outcome on mesophyll differentiation[7]. Overexpression of *EPF1* or its close relative *EPF2* in Arabidopsis has been shown to lead to a decrease in stomatal density[25] and overexpression of a cognate gene in wheat (*TaEPF1*) has recently been shown to lead to a similar phenotype[26].

Confocal imaging of the TaEPF1-OE lines revealed that some cells in the stomata-forming epidermal files appear to have undergone the initial events of stomata formation but failed to undergo the final division process to generate the stomatal complex and pore (Fig. 3a). The mesophyll cells directly subtending these abnormal stomatal-progenitors showed no sign of cell separation, whereas mature stomata in displayed clear sub-stomatal cavities (Fig. 3b, c). Counting the presence/absence of sub-stomatal cavities confirmed a complete lack of airspace cavities below the aborted stomatal lineage cells in the TaEPF1-OE wheat lines, whereas all differentiated stomata were subtended by cavities (Fig. 3h). MicroCT imaging of the TaEPF1-OE leaves also revealed a lack of sub-stomatal cavities compared to WT (Fig. 3d, e) and a denser mesophyll (Fig. 3f, g). Quantification of

leaf structure revealed the TaEPF1-OE leaves did indeed have a lower porosity than WT (Fig. 3i). Measurements of gas exchange revealed a significant decrease in $g_s$ in the TaEPF1-OE lines compared to control non-transgenic leaves (Fig. 3j).

To further investigate the potential linkage of sub-epidermal cell separation with the events of stomatal differentiation, we exploited the fact that the longitudinal axis of a grass leaf provides a developmental gradient of tissue differentiation, with cells at the proximal base undergoing division to generate cells which enter a number of developmental pathways, including the formation of stomata, at more distal tip regions[23]. Analysis of the control line (the untransformed parent of the transgenic TaEPF1-OE plants) did not reveal any gradient in stomatal density in the regions observed at the tip, middle or base of mature leaf 5 (Supplementary Fig. 4a). However, a similar analysis of comparable leaves of the transgenic TaEPF1-OE lines indicated a decrease in stomatal density at the base of leaf 5 (Supplementary Fig. 4b) which was reflected by a decrease in porosity in this region, as revealed by CT analysis (Supplementary Fig. 4c). To better identify regions at the base of wheat leaves in which stomatal differentiation was just starting, we used confocal microscopy to analyse leaf 3 of wheat seedlings at a relatively early stage of development. This revealed that the more distal tip regions of these leaves were distinguished by having mature stomatal complexes (Fig. 4a) subtended by relatively large airspaces (Fig. 4b, c). In contrast, in the more proximal base regions where epidermal cells undergoing division patterns characteristic of stomatal differentiation were clearly visible (Fig. 4d), although occasional airspaces were visible at the interstices of some adjoining sub-epidermal cells (Fig. 4e), these were both much smaller than those observed in the more proximal tissue (Fig. 4b) and did not show any overt pattern linked to the overlying differentiating stomata (Fig. 4f).

These results support the hypotheses that cell separation occurs in the sub-epidermal mesophyll as part of an endogenous programme of development, but that the size and distribution of the airspaces eventually formed is promoted by the presence of adjacent, differentiated stomata. The data suggest that there is a causal link between stomatal density and the overall porosity of the mesophyll, and that gas exchange and mesophyll porosity are functionally linked.

To investigate potential physiological drivers for these changes, we compared the photosynthetic assimilation rate, mesophyll conductance to $CO_2$ ($g_m$), and instantaneous water use efficiency (iWUE) of the wheat lines with different ploidy levels. This revealed no overt trend linking assimilation rate (Supplementary Fig. 5a) or $g_m$ (Supplementary Fig. 5c) with ploidy level, and no obvious correlation of assimilation rate with mesophyll porosity (Supplementary Fig. 5e). In contrast, the 6n wheat lines had a significantly higher iWUE than the 2n and 4n lines (Supplementary Fig. 5b; ANOVA with posthoc Tukey, $P = 0.0005$ and $P = 0.013$, respectively). This was mirrored by a decrease in exposed mesophyll surface area per volume calculated from our CT analyses (Supplementary Fig. 5d), with 6n lines having significantly lower values compared with the 2n lines (ANOVA with posthoc Tukey, $P = 0.0001$). Interestingly, measurements of mesophyll cell volume via confocal light microscopy revealed a clear increase with ploidy level (Supplementary Fig. 5f). Bearing in mind the fixed relationship of surface area to volume for similar shapes, these data fit with a hypothesis that during the selection of modern wheat there has been an increase in mesophyll cell size, with concomitant decrease in exposed surface area and decrease in mesophyll porosity, as well as altered stomatal parameters of increased size and decreased density. How these are mechanistically linked and integrated at the whole leaf level to improved water use efficiency awaits further research.

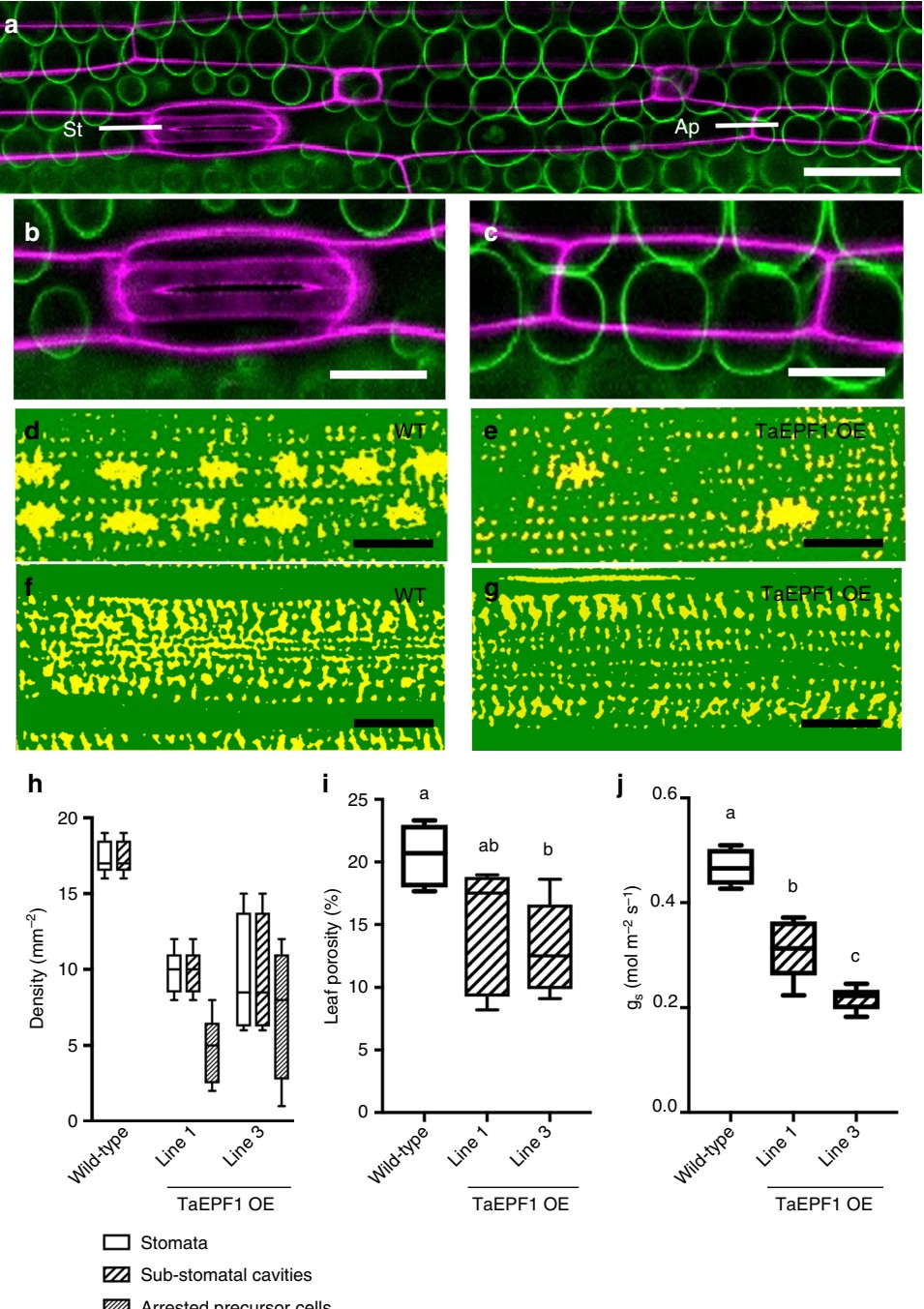

**Fig. 3** EPF overexpression arrests sub-stomatal cavity development and decreases mesophyll porosity in wheat. **a** Confocal overview of a TaEPF1 OE wheat leaf showing epidermal layer (purple), subtending mesophyll cells (green), a stomate (St) consisting of guard cells and associated subsidiary cells and, in the same file, an arrested stomatal precursor (Ap). Scale bar = 60 μm. **b**, **c** Higher resolution images of (**b**) the stomate and (**c**) arrested stomatal precursor cell shown in **a**. Scale bars = 40 μm. **d**–**g** microCT images of a wild-type (WT) (**d**, **f**) and a TaEPF1 OE leaf (**e**, **g**) in a paradermal plane within the mesophyll directly subtending the epidermis (**d**, **e**) or deeper in the leaf (**f**, **g**), with solid tissue in green and airspace in yellow. The larger airspaces in **d**, **e** indicate sub-stomatal cavities. Note fewer sub-stomatal cavities in the TaEPF1 OE leaf. Scale bars = 100 μm. **h**–**j** In WT wheat and two independent lines of transgenic wheat overexpressing TaEPF1 (as indicated), (**h**) the density of stomata ($n = 87$), sub-stomatal cavities ($n = 87$) and arrested stomata precursor cells ($n = 52$ from 5 independent leaves); (**i**) mesophyll porosity as measured from microCT analysis (ANOVA, $F_{(2,12)} = 4.977$, $p = 0.027$); and (**j**) stomatal conductance, $g_s$, are shown (ANOVA, $F_{(2,12)} = 46.86$, $p < 0.0001$). For **i** and **j** a posthoc Tukey analysis was performed ($n = 5$). Lines sharing the same letter are indistinguishable from each other at the $p < 0.05$ confidence limit. Data (**h**–**j**) are shown as box plots (25th–75th percentile, horizontal line = median) with whiskers indicating maximum and minimum values

To investigate the potential causal linkage between stomatal differentiation and mesophyll airspace formation, we turned our analyses to eudicots, subjecting a series of Arabidopsis transgenic lines previously shown to have altered stomata density[27] to a combined microCT and gas exchange analysis. We focused on lines in which overexpression of *EPF2* leads to leaves with a significantly lower stomatal density (*EPF2OE*), and where loss of *EPF2* and its homologue (*EPF1*) (*epf1epf2* mutant) generates

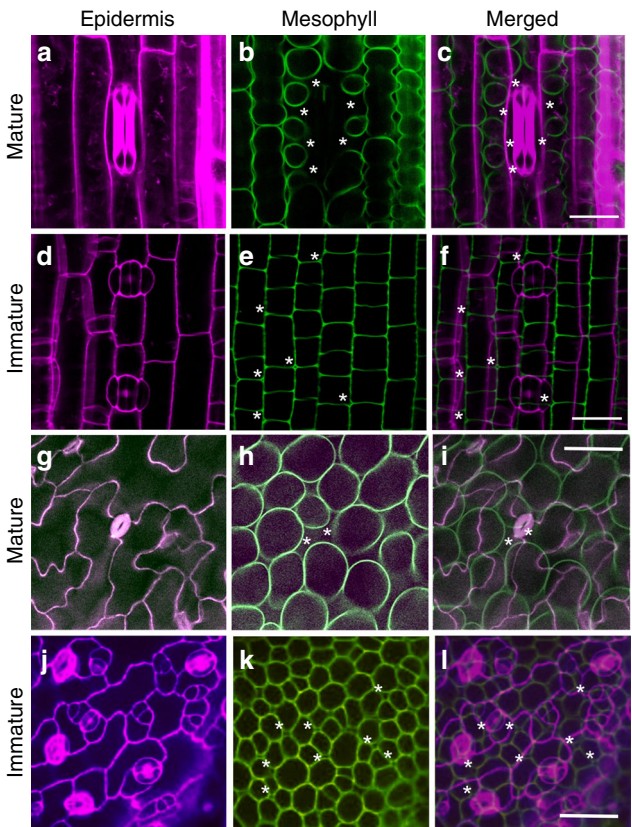

| Epidermis | Mesophyll | Merged |

**Fig. 4** Developmental progression of stomatal differentiation and mesophyll airspace formation. **a–f** Confocal images of leaf 3 of wheat (6n) seedlings taken either at the distal tip region (**a–c**) or proximal base (**d–f**). Images are from the epidermis (**a**, **d**) or subtending mesophyll (**b**, **e**), with cell walls false-coloured and overlaid in (**c**, **f**). A mature stomata is visible in **a**, with two immature stomata in **d**. A large airspace subtends the stomata in **a** (indicated by asterisks in **b**, **c**). Small airspaces (asterisks) are visible in **e**, **f** at cell junctions. **g–l** Confocal images of Arabidopsis leaves at maturity (**g–i**) or early development (**j–l**). Images are from the epidermis (**g**, **j**) or subtending mesophyll (**h**, **k**) with cell wall false-coloured and overlaid in **i**, **l**. A mature stomata is visible in centre (**g**), with numerous stomata at various developmental stages in **j**. A relatively large airspace (asterisks) is visible below the central stomate (**h**), whereas some very small airspaces (asterisks) are distributed within the immature mesophyll (**k**, **l**) at cell junctions. Scale bar **c**, **f** = 20 μm; **i**, **l** = 25 μm

leaves with a significantly higher stomatal density[25]. Exemplar microCT images for each line are shown in Fig. 5a, b, d with SEM images of stomata shown in Fig. 5e, f, h and confirmation of the expected stomatal density phenotype shown in Fig. 5i.

Eudicot leaves (as found in Arabidopsis) are distinct from typical monocot leaves (exemplified by wheat in this paper) in that they characteristically show two mesophyll regions, the adaxial palisade and abaxial spongy layers, which are distinguishable by having developmentally set differences in cell shape and airspace (porosity)[13]. By selecting volumes of interest within leaves of the Arabidopsis lines, we obtained distinct porosity data for the respective palisade and spongy layers. Although comparison of mean overall mesophyll porosity values suggested limited differences between the lines, this mainly reflected the similarity in values for spongy mesophyll porosity (Supplementary Fig. 6). In contrast, the palisade mesophyll porosity (which is more strongly associated with photosynthetic efficiency than spongy mesophyll porosity[13]) showed greater differences between the lines, with the *epf1epf2* palisade having the highest porosity

(Fig. 4j), and the highest stomatal density (Fig. 4i). These data suggest a more complicated situation in the eudicot Arabidopsis leaf compared to the monocot wheat leaf. Manipulations that alter stomatal density in Arabidopsis can lead to overt alterations in mesophyll porosity, but the outcome is dictated by the identity of the mesophyll (palisade or spongy). The observations fit to an interpretation whereby a developmental pattern of mesophyll differentiation is set very early in leaf development (defining palisade and spongy layers in eudicots)[28,29] with mesophyll identity setting the scale of modulation of porosity by factors that occur later in development, such as signals associated with stomatal patterning. Furthermore, as with wheat, an analysis of early stages of stomatal differentiation supported the idea that the degree and extent of mesophyll airspace formation was modulated by the presence of mature, differentiated stomata. Thus, during the phase of leaf growth in which the epidermal cell divisions leading to stomata were occurring (Fig. 4j), airspaces were visible at the interstices of some sub-epidermal cells (Fig. 4k), but there was no overt similarity in the pattern of these airspaces and the overlying differentiating stomata (Fig. 4l). By the stage of development in which fully differentiated stomata were visible (Fig. 4g), there were numerous, large airspaces throughout the sub-epidermal palisade mesophyll (Fig. 4h), with stomata always subtended by an airspace (Fig. 4i).

**Stomatal conductance and modulation of mesophyll airspace.** Our data from both wheat and Arabidopsis support the hypothesis that the presence of stomata modulates the degree of mesophyll porosity. However, our observation that palisade porosity in the *EPF2OE* line in Arabidopsis is not different from *Col-0* (Fig. 5j), despite a massive decrease in stomatal density (Fig. 5i), suggests that it is not simply a direct process in which differentiation of guard cells leads to a proportionate increase/decrease in mesophyll cell separation (as has been suggested[7]). An alternative (though not exclusive) hypothesis is that the actual functioning of stomata to allow gas exchange is a major driver linking stomatal density and mesophyll porosity. Indeed, our wheat data show a strong positive correlation between mesophyll porosity and $g_s$ (Figs. 1l, 3i, j). We tested this hypothesis in Arabidopsis by exploiting a recently characterised stomatal mutant, *focl1-1*, in which the final steps of guard cuticular ledge formation are disrupted[30]. This results in most of the stomata forming pores that are initially completely covered over by a lipid/cuticle layer. As the they mature the cuticle across approximately 10% of stomata becomes torn, leading inevitably to holes which must allow a limited degree of gas exchange into and out of the leaf via the visible sub-stomatal cavities formed. A partially covered *focl1-1* stomate is shown in Fig. 5g and can be compared with the open stomatal pores observed in *Col-0*, *EPF2OE*, and *epf1epf2* leaves (Fig. 5e–h). The presence of partially covered stomatal pores provides a useful tool to investigate the extent to which mesophyll porosity is linked to gas exchange. A pairwise combined microCT and gas exchange analysis of the four Arabidopsis lines revealed a highly significant positive correlation of $g_s$ and palisade porosity (Pearson correlation, $r^2 = 0.471$, $P = 0.0002$; Fig. 5l), with the *focl1-1* mutant data grouping below those for the *Col-0* control and on a par with the *EPF2OE* samples. When assimilation rate for individual leaves of the mutant plants is considered with respect to palisade porosity, although there is a weak correlation (Supplementary Fig. 7a; Pearson correlation, $r^2 = 0.289$, $P = 0.007$), the relationship of porosity to water-use efficiency is much stronger (Supplementary Fig. 7b; Pearson correlation, $r^2 = 0.526$, $P = 0.0001$), consistent with the hypothesis that restricting water loss may be the primary driver for the changes in leaf structure observed.

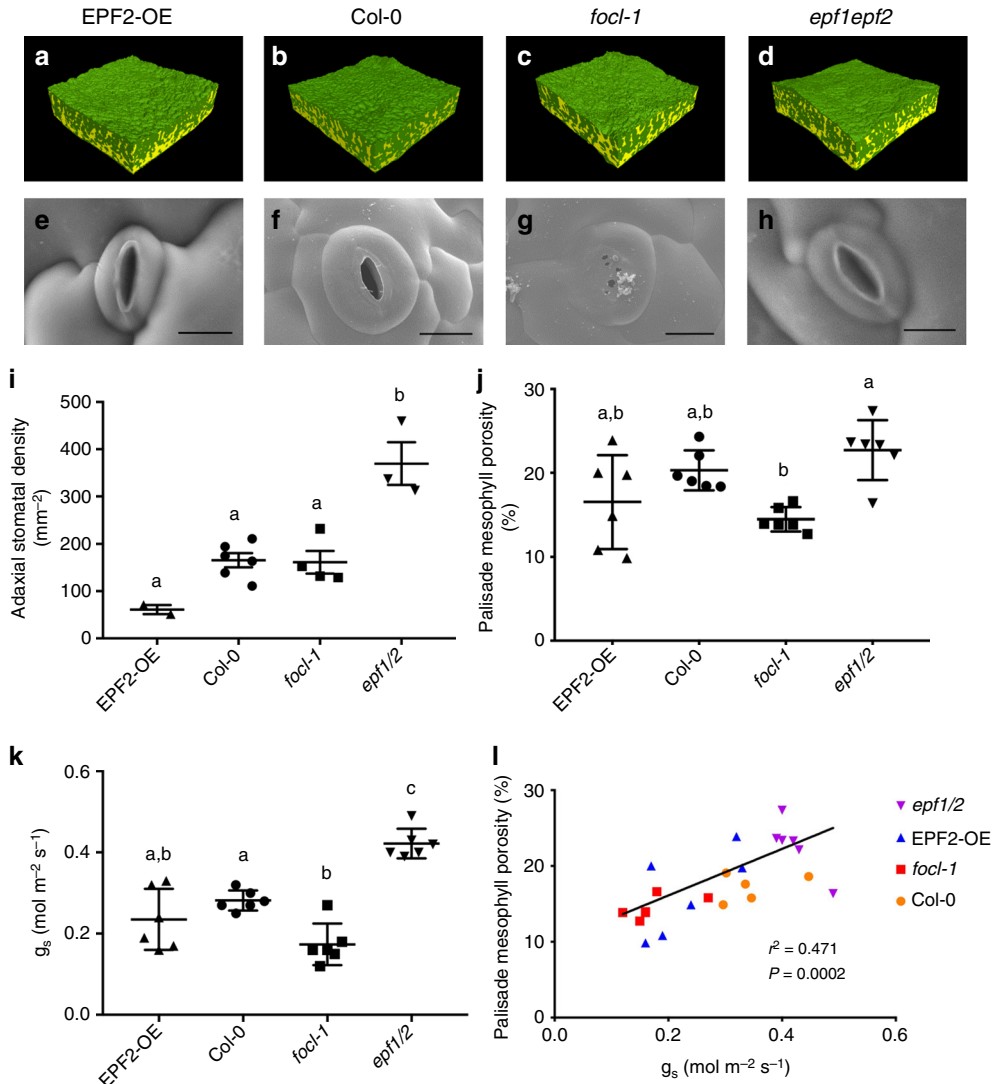

**Fig. 5** Mesophyll porosity is modulated by gas exchange through stomatal pores. **a–d** 3D microCT renderings of tissue blocks (resolution = 2.75 μm; samples = 1.1 mm²) and (**e–h**) exemplar SEM images of stomata from leaves of Arabidopsis *EPF2-OE*, *Col-0*, *focl1-1* and *epf1epf2* lines (scale bars = 10 μm). Means and standard deviation are shown for (**i**) stomatal density (ANOVA, $F_{(3,11)} = 19.17$, $p < 0.0001$), (**j**) palisade mesophyll porosity (ANOVA, $F_{(3,20)} = 6.329$, $p = 0.0034$) and (**k**) stomatal conductance, $g_s$ (ANOVA, $F_{(3,20)} = 26.22$, $p < 0.0001$) for the Arabidopsis lines in **a–d**, with $n = 6$ except for stomatal density where $n = 4$ (*focl-1*), $n = 2$ (*EPF2-OE*) and $n = 3$ (*epf1epf2*). Col-0 data are as in ref. [13]. Lines indicated with the same letter cannot be distinguished from each other at the $p < 0.05$ confidence limit (posthoc Tukey). **l** Palisade mesophyll porosity is plotted against stomatal conductance, $g_s$, for individual leaf samples from the four Arabidopsis lines, as indicated. Linear regression results are presented

Overall, these observations support the idea that, in addition to a potential direct signal from differentiated guard cells, the actual functioning of stomata to allow gas exchange plays an important functional role in promoting the cell separation and growth events which ultimately control mesophyll porosity.

## Discussion

Taken together, our data support the hypothesis that the degree and extent of separation of mesophyll cells to form airspaces is linked to the presence of functional pores (i.e., stomata that allow gas flux), rather than simply relying on the presence of differentiated guard cells to generate an (as yet) unspecified direct signal. The existence of a conductance-based mesophyll patterning system that would allow the leaf to adjust mesophyll differentiation to its actual photosynthetic functioning is appealing since it would permit leaf anatomical specialisation to match the growth environment, as is observed[17]. The nature of

such a flux-signal is open to debate. One possibility is that the concentration of $CO_2$ to which the mesophyll is exposed influences porosity. Alternatively, exposure to the atmosphere will influence the local vapour pressure deficit around the mesophyll cells, thus it is possible that some element of water vapour loss is linked to the degree of cell separation events that occur. These questions, as well as the nature of the molecular events coordinating cell separation and growth within the leaf, remain to be resolved. Irrespective of this, our data establish a plausible mechanism linking stomatal function and mesophyll airspace formation in both eudicots and monocots.

In addition to this advance in our fundamental understanding of leaf differentiation, our data provide an insight into wheat domestication. Stomata play a key role in controlling water use in crops and we reveal that during the evolution and domestication of modern bread wheat (a staple food crop for the world) there has been a step-wise selection for leaves with a decreased stomatal density/increased stomatal size, which is associated with both a

decrease in $g_s$ and a decrease in mesophyll porosity (i.e., to yield a denser leaf). We propose that there is a functional relationship between $g_s$ and mesophyll differentiation and that this relationship has underpinned at least some of the selection process by which modern crops show improved water use efficiency.

## Methods

**Plant material and growth conditions.** Seeds of *Triticum baeoticum* (2n) (TRI 18344), *T. urartu* (2n) (TRI 6735), *T. araraticum* (4n) (TRI 18513), *T. dicoccoides* (4n) (TRI 18465) (obtained from IPK Gatersleben, Germany (https://www.ipk-gatersleben.de/en/gbisipk-gaterslebendegbis-i/)), three cultivars of *T. aestivum* (6n) (cv. cougar, crusoe and shango) (provided by RAGT seeds, Cambridge, UK), and the wheat TaEPF1-OE1 line[26] were germinated in 6:1 Levington M3: Perlite and grown in a controlled environment chamber (Conviron, Manitoba, Canada) with 16 h photoperiod set to 21 °C, 60% relative humidity (RH), and 400 μmol m$^2$ s$^{-1}$ at bench level (425 μmol m$^2$ s$^{-1}$ at the top of the canopy) photosynthetic photon flux density (PPFD). The youngest fully expanded leaves of 30-day-old plants (leaf 5 or 6) were used for stomatal conductance and microCT analysis.

For *Arabidopsis thaliana* lines Col-0, *EPF2OE*, *epf1epf2*, and *focl1-1* (generated and maintained in our group[30,31]) seeds were stratified on wetted filter paper or directly on compost at 4 °C in the dark for 7 days, then germinated on M3 compost. All plants were grown in a Conviron MTPS 120 growth room (Manitoba, Canada) with a 12 h photoperiod set to 21/15 °C day/night temperatures, 60% RH, and 200 μmol m$^{-2}$ s$^{-1}$ PPFD at canopy height. The largest mature leaf (fully expanded but with no senescence) was used for stomatal conductance and subsequent microCT analysis, as described below. Leaves were sampled between 34 and 38 days after sowing.

**Gas exchange.** Leaf level physiology was measured in both wheat and Arabidopsis lines using a LI-6400XT portable IRGA coupled with 2 cm$^2$ Li-6400-40 leaf chamber fluorometer (LI-COR, Lincoln, NE, USA). Spot measurements of stomatal conductance to water vapour ($g_s$) and net assimilation rate ($A_{400}$) were collected on four replicate wheat plants and six replicate Arabidopsis plants, and used to calculate instantaneous water use efficiency (iWUE) as $A_{400}/g_s$. For wheat lines, chamber conditions were controlled at 24 °C leaf temperature, a saturating light level of 1500 μmol m$^{-2}$ s$^{-1}$, reference $CO_2$ concentration of 400 μL L$^{-1}$, and approximately 60% RH. For Arabidopsis lines, chamber conditions were controlled at 21 °C leaf temperature, a saturating light level of 1200 μmol m$^{-2}$ s$^{-1}$, reference $CO_2$ concentration of 400 μL L$^{-1}$, and approximately 60% RH. Leaves were pre-acclimated to these chamber conditions until reaching steady state and full stomatal opening (i.e., ~20–30 min) before $g_s$ was logged for each plant. Measurements were collected on the youngest fully expanded leaf of 30-day-old wheat plants and the largest leaf of 35-day-old Arabidopsis plants. Mesophyll conductance to $CO_2$ ($g_m$) was calculated[32] using the LandFlux fitting tool (version 10.0; http://landflux.org/Tools.php) using a series of assimilation data points collected across a range of $CO_2$ concentrations under the same light and leaf temperature conditions described above.

**X-ray computed microtomography (CT) imaging and analysis.** The same leaf used for $g_s$ measurements was then imaged using an X-ray micro CT scanner (Nanotom, General Electric Company, USA) as a 5 mm diameter leaf disc cut from one side of the mid-rib for Arabidopsis and as a 1 cm$^2$ section of wheat leaf blades, again avoiding the mid-rib. Samples were kept static during image acquisition to maximise image quality by placing between two pieces of polystyrene. Imaging of leaf discs was undertaken at 2.75 μm of spatial resolution, with energy of 65 kV and 140 μA current, and collecting 2400 projections with an exposure time of 750 ms, resulting in total scan time of 30 min per leaf disc.

2D projections (radiographs) acquired during the scans were reconstructed (Datos|X; General Electric Company, USA) into 3D volumes using a filtered back-projection algorithm, rendered, and converted into stacks of Tiff images (VG Studio Max version 2.2; Volume Graphics, Germany). Image stacks were used to create a mask separating the leaf disc from the background (Avizo version 6.0; FEI, USA). Image stacks with leaf discs separated from the background were cropped to remove any damaged area on the edges. Cropped and aligned image stacks of leaf discs were thresholded in ImageJ software[33] using IsoData and minimum algorithms to obtain stacks of images representing the leaf disc mask and plant material only. It was necessary to occasionally swap between these two algorithms between scans from different days due to variations in brightness caused by filament decay. The generated image stacks were used as the input to the ImageJ image calculator and the function XOR was used to generate image stacks with only pore space visible. As the last stage of image analysis, the structural descriptors of intercellular pore space including porosity and mesophyll surface area were obtained using the ImageJ particle analyser and BoneJ[34].

Where separate spongy and palisade mesophyll values are reported in Arabidopsis, these are based on a single representative slice. Despite aligning the leaves horizontally, there were always a very few slices that were clearly composed of only one mesophyll tissue type and entirely within the masked volume. Representative palisade slices were selected by visual inspection according to these

criteria. For the spongy mesophyll, the slice with the maximum porosity value was selected since plots of porosity over the depth of the leaf consistently showed a pronounced peak in the spongy mesophyll (e.g., Fig. 2m–o).

**Light and electron microscopy.** Confocal microscopy of wheat leaves was adapted from[35]. An approximately 1 cm$^2$ piece of tissue from the middle third of a leaf blade was fixed using 3:1 ethanol: acetic acid (v/v) containing 0.05% Tween 20. After 1 h of vacuum infiltration, samples were stored at 4 °C for at least 48 h. Samples were rinsed in 50% ethanol/water for 5 min, followed by 5 min in 70% ethanol before a 15-min chloroform rinse. Samples were then rehydrated step-wise through 70% ethanol, 50% ethanol, water (5 min per wash), then bleached in 0.2M NaOH containing 1% (w/v) SDS. After washing with water, samples were treated with 0.01% (w/v) alpha-amylase in PBT (PBS + 0.1% Tween-20) at 37 °C overnight to remove starch. Samples were then treated with freshly prepared 1% (w/v) periodic acid for 40 min before being washed with water, then stained for 4 h in the dark at 4 °C with freshly prepared pseudo-schiff propidium iodide (0.1 M Na$_2$S$_2$0$_5$, 0.15N HCL, 0.01% (w/v) propidium iodide). Samples were rinsed 3 times in water, left overnight water at 4 °C, then cleared for at least 6 h in a few drops of chloral hydrate in glycerol/water before being mounted on slides using 20% (v/v) Arabic gum in chloral hydrate/glycerol. Samples were kept at 4 °C in the dark prior to imaging using an Olympus FV1000 confocal microscope.

Imaging of stomata using scanning electron microscopy was performed on both a Zeiss EVO HD15 SEM fitted with a cryo-stage[30] and on a Hitachi TM3030plus SEM. For the former, frozen samples were sputter coated with platinum to a depth of 5 nm, then images captured with a gun voltage of 6 kV, I probe size of 460 pA at a working distance of 5 to 6 mm with a secondary electron detector. For the latter, leaf samples were mounted with cryogel on aluminium stubs, placed on a cooling stage and rapidly frozen to −20 °C prior to imaging under vacuum at 15 kV with secondary electron detection.

**Stomatal density and maximum anatomical stomatal conductance.** Stomatal counts in wheat were made by applying dental resin (Coltene Whaledent, Switzerland) to the epidermis of the leaf blade. Once this had set, it was removed from the leaf and a coat of clear nail varnish was applied to the resin. Stomatal density counts were made from these nail varnish impressions via light microscopy (Olympus BX51; Tokyo, Japan). Five counts were made either side of a major vein. This was repeated for both the abaxial and adaxial surfaces of the leaf and summed to show total stomatal density.

Anatomical maximum stomatal conductance ($g_{smax}$), and measurements of stomatal size, were made from sections of mature wheat leaves from six replicate plants per line (Supplementary Fig. 3d). Leaf sections were prepared following[35], with the exception that samples were stained in pseudo-schiff propidium iodide for only 4h, then imaged via confocal laser scanning microscopy (Olympus Fv1000; Tokyo, Japan) and light microscopy (×20 objective; Nikon BX51; Tokyo, Japan). All samples were imaged within 48-h of mounting. Anatomical $g_{smax}$ for imaged leaves was calculated following equation (1)[7,36].

$$\text{Anatomical } g_{smax} = d.\, D.\, a_{max}/(v.\,(l+(\tfrac{\pi}{2}).\sqrt{(a_{max}/\pi)})) \tag{1}$$

In this equation, $d$ is the diffusivity of water in air at 22 °C (m$^2$ s$^{-1}$), $v$ is the molar volume of air at 22 °C (m$^3$ mol$^{-3}$) and $\pi$ is a mathematical constant (approximated to 3.142). $D$ is stomatal density (mm$^{-2}$), $a_{max}$ is mean maximum stomatal area (μm$^2$) and $l$ is pore depth (μm). $l$ is assumed to be equivalent to guard cell width at the middle of the stoma. Values were estimated for the abaxial and adaxial leaf surfaces and summed to calculate total anatomical $g_{smax}$. Values of $D$ from individual leaves were used in calculations, whilst averages for each line were used for $a_{max}$ and $l$.

**Reporting summary.** Further information on research design is available in the Nature Research Reporting Summary linked to this article.

## Data availability

The datasets generated during the current study are available from the corresponding author on reasonable request. The source data underlying Figs. 1g-l, 2m-o, 3h-j and 5i-l and Supplementary Figs. 2, 3a-d, 4a-c, 5a-f, 6a,b and 7a,b are provided as a Source Data file.

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

## Acknowledgements
The project was funded by BBSRC grant BB/J004065/1 'Optimising photosynthetic efficiency via leaf structure' to A.J.F., S.M., J.G., C.P.O., C.S., and S.R. J.D. was funded by a BBSRC WR DTP studentship to J.E.G. and A.J.F.; M.W. was funded by a BBSRC iCASE studentship to A.J.F. and C.P.O. (with RAGT Seeds, Cambridge, UK); A.L.B. was funded by a Gatsby Foundation PhD studentship.

## Author contributions
M.R.L., R.P., J.D., A.L.B., M.J.W., A.M., L.H., M. F.-S. performed the experiments; C.J.S. and S.J.M. supervised the CT analysis and interpretation; S.R. and C.P.O. advised on fluorescence-linked gas exchange analysis and interpretation; J.E.G. led and supervised the wheat EPF transgenic studies and advised on the Arabidopsis stomatal mutant analysis; M.R.L., A.L.B. and A.J.F. interpreted the results and wrote the paper, with contributions from all authors. A.J.F. designed the study and led the project.

## Additional information

**Competing interests:** The authors declare no competing interests.

