## [Peer Review File · Nature Communications]

Reviewers' comments:

Reviewer #1 (Remarks to the Author):

This is an interesting study, combining different lines of evidence supporting the idea that the presence of functional stomata leads to the formation of mesophyll air spaces during leaf development. My only concern is on the validity of the porosity – stomatal conductance (g_s) relationships, which is the basis of the conclusions. This is because the strong relationship shown in Fig. 1I only holds when 'average' values are considered (and no deviation is shown in the graph). But looking at g_s data in fig. 1K it appears that there is a lot of variability within each genotype. For instance, in hexaploid Shango cv, g_s can be wherever between 0.3 and 0.55, i.e. at any point along the g_s axis in Fig. 1I. The way to produce really significant test for the regression is to use individual data rather than average data per genotype. That is, take a leaf and measure g_s , immediately after sample that same leaf to be analysed for porosity, and provide paired data relationship to test whether the correlation shown really holds. As another minor issue, up to three different gas exchange analysers were used, yet it is unclear in Methods section which one was used on each experiment.

Reviewer #2 (Remarks to the Author):

Lundgren *et al*. reported an interesting study on substomatal cavity formation. When formation of stomatal complex was initiated but not completed in the mutant lines, substomatal cavity was not formed. This implies that the complete stomatal complex formation is prerequisite to the substomatal cavity formation. However, the authors did not follow the developmental processes of the stomatal complex or those of substomatal cavity. Because there is an age gradient along the grass leaf, the authors easily compare these developmental processes. Then, which is the first can be known.

In *Arabidopsis thaliana* plants, porosity of the palisade tissue was analysed, which is plausible. However, the stomatal conductance was measured for the whole leaf. Because the contribution of the stomata on the abaxial side of the leaf would be great, this reviewer wonders whether the authors are able to evaluate the stomatal conductance of the adaxial side. The stomatal conductance was measured in mature leaves. Thus, again, the sequence of the events is not clear.

This reviewer suggests that the authors should add some developmental data for wheat as well as for *A. thaliana*. This will clarify the sequence of the events.

2) For *A. thaliana*, the adaxial stomatal data may be examined. The morphological estimate of stomatal conductance may be sufficient for this purpose.

Minor points

L. 94: It is necessary to define width, area and length of stomatal complex. Perhaps, it is easy to use an illustration like the complex shown in Figure 1 d.

L. 113: wider stomatal complex? greater stomatal complex??

Figures: P may be in italic rather than in lower-case roman.

Figure S2: mesophyll surface area may be expressed on leaf surface area.

For Figure S2-a(b), which is the explanatory variables g_s or mesophyll surface (porosity)?

Reviewer #3 (Remarks to the Author):

This is an interesting paper that addresses the link between stomata and the underlying air-spaces in the mesophyll, a question that has important physiological and developmental relevance. The underlying concept is that because CO₂ must traverse the epidermis and the mesophyll air spaces in series, one might expect that their permeabilities are developmentally coordinated. The specific question addressed is whether it is the presence of stomata or their opening (and thus resulting gas exchange) that triggers air space development.

The study uses plant material with varied stomatal characteristics to show that there is a link between stomata and air-space development. The first figure uses different species and varieties of wheat (with varied ploidy) to demonstrate a correlation between *g_s* and mesophyll porosity. The transgenic OE of EPF1 in wheat was quite convincing: when stomata fail to differentiate, substomatal cavities do not form below them. However, (and as the authors' note) this could be due to a signal from the stomata or from their opening.

It is the Arabidopsis studies that the authors use to support the idea that it is gas exchange through stomata, and not just their presence, that is responsible for the linkage. Here the relation between stomatal conductance and air space porosity occurs only in the palisade mesophyll, and while the data are consistent with the authors' conclusions and indeed quite interesting, I wish that the paper did more to demonstrate a role for gas exchange in coordinating air space formation.

Specifically, the paper would be enhanced by clarification of the timing of the developmental and physiological events. With reference to % leaf expansion – when do stomata first become competent to open and when do the mesophyll cells separate? Could the authors do more to place these processes in a developmental context? Do we know that the four Arabidopsis lines compared in Figure 3 all followed the same patterns of leaf expansion and final size, thickness, etc.? I also wondered why there were no experimental manipulations? It seemed to me that the system would be ripe for physiological experimentation and that this would help the paper arrive at a more convincing understanding of how such a “conductance-based mesophyll patterning system” might indeed operate.

Reviewer #4 (Remarks to the Author):

This manuscript addressed an important developmental question in plant biology that has huge implications for plant physiological function, whether plants coordinate cuticular gas exchange with mesophyll porosity.

I liked the approach employed by the authors utilizing two species with very different leaf anatomies as well as mutants and the novel method of MicroCT to visualize the anatomy of the internal leaf.

Major comments:

- How can we rule out changes in cell size due to the ploidy changes driving the calculated porosity changes of the mesophyll in wheat? There seems to be a huge variation from the MicroCT scans of cell size that to my eye correlates with porosity. Cell size was not quantified and analyzed and should be, considering it is driven by genome size changes.
- Given the importance of mesophyll porosity for photosynthesis, please show the effect of these changes in porosity on photosynthesis. If these changes in mesophyll porosity are important there should be large differences in photosynthetic rate.
- The authors constantly refer to porosity as an important driver of photosynthesis and gas exchange properties of the leaf yet never measured mesophyll conductance to CO₂? I imagine also that if mesophyll porosity as determined in this manner is important for photosynthesis then we ought to see a difference in mesophyll conductance. The authors have all of the equipment to measure this important trait.
- The TaEPF1-OE lines were an excellent system in which to show the presence of a fully developed

stomata determines the formation of a substomatal cavity. Image d of Figure 3 is however at a very different plain to figure 2. Why the huge difference? There is an obvious effect of a stomata on the presence of a substomatal cavity but how does that relate to mesophyll porosity as determined in Figure 2 jkl? In figure 2abc there is no layer clearly shown in these renditions that depicts the layer shown in Figure 3d. Yes I agree that stomata=substomatal cavity (a great result) but what about the mesophyll at the same height as calculated in Figure 2. This is an inconsistency across this manuscript that porosity seems to be determined from a non-standard paradermal layer, either immediately below the epidermis (which would then just be a function of the number of stomata, an unsurprising result given sub-stomatal cavity is named for its known presence below stomata) and then in Figure 2 in the middle (??) of the leaf any correlation could then be due to simple changes in cell size but not leaf thickness. In addressing this concern please add images at the same plain of Figure 2 jkl for those of Figure 3 d and e.

Concerns with the method:

-Given that sections of transpiring leaves were taken for imaging under the MicroCT, how can we be certain that these differences in calculated mesophyll porosity are not due to changes in mesophyll cell size due to a drop in bulk cell turgor that happens when leaves are transpiring and under more negative water potentials? Just because the leaf section was enclosed in polystyrene doesn't mean that when the sample was excised the leaves didn't have different turgors due to different gs. I think that there might be very different results for non-transpiring, fully hydrated leaves, in this state we would get the true measure of comparative developmental differences in mesophyll anatomy. There could be an auto-correlation between mesophyll porosity in the higher transpiring leaves because they would have more negative water potentials, cell turgors and thus smaller mesophyll cells and greater porosity. There is lots of literature using cryosem showing that there are massive changes in mesophyll cell shape as leaf water potential declines, even slightly.

Minor comments:

Line 32: 'evolutionary significance' is a vague statement, considering in the same sentence the coordination of intercellular spaces and stomata is said to be essential for photosynthesis.

Line 109-115: to me this is an overreach - change like Line 143 to indirect. Genome size correlates with cell size so how can the authors claim that there has been active selection for cultivars with larger stomata. Isn't it more likely that domestication favoured larger cell sizes due to yield increases?

Figures: Figure 2 Please mark with a dotted line on jkl where the transverse and longitudinal sections/slices were taken from and visa versa.

Line 190: universality is a strong word - even with the caveat of potentially, considering only 2 angiosperm species were used. Note the work by Duckett and Presel in mosses and intercellular air spaces, this work should be consulted and referred to esp with regards to Lines 240-242.

Figure 4. Does the focl-1 mutant have a substomatal cavity? Please check and discuss.

Figure 4I, again I would like to see how this difference in porosity relates to photosynthesis if that is why this is important.

Lundgren et al Response to Referees

Reviewer #1 (Remarks to the Author):

This is an interesting study, combining different lines of evidence supporting the idea that the presence of functional stomata leads to the formation of mesophyll air spaces during leaf development. My only concern is on the validity of the porosity – stomatal conductance (g_s) relationships, which is the basis of the conclusions. This is because the strong relationship shown in Fig. 1I only holds when ‘average’ values are considered (and no deviation is shown in the graph). But looking at g_s data in fig. 1K it appears that there is a lot of variability within each genotype. For instance, in hexaploid Shango cv, g_s can be wherever between 0.3 and 0.55, i.e. at any point along the g_s axis in Fig. 1I. The way to produce really significant test for the regression is to use individual data rather than average data per genotype. That is, take a leaf and measure g_s , immediately after sample that same leaf to be analysed for porosity, and provide paired data relationship to test whether the correlation shown really holds.

We have performed the paired data analysis as requested by the reviewer. The results support the conclusions made in the original figure part 1I, with an r^2 value of 0.451 at a P value = 0.0001, i.e., there is a strong relationship of g_s and mesophyll porosity across the different ploidy levels in wheat. At present we provide this analysis as a new supplementary figure (Fig S2) with text (lines 151-153) since we believe the original version (Fig. 1I) is easier for the non-specialist to interpret, but we would be perfectly happy to switch this around if the reviewer/editor so wish.

As another minor issue, up to three different gas exchange analysers were used, yet it is unclear in Methods section which one was used on each experiment.

We now present all data collected using the same gas exchange analyser, the LI-COR 6400XT photosynthesis machine and this is now described in the Methods section (lines 337-353). We apologise for any previous confusion. As a consequence, to ensure absolute comparability of data, some of the gas exchange (and corresponding paired microCT) data are new, including additional use of Col-0 results from Lehmeier et al (2017) Plant J 92, 981-94 as reference data. Our conclusions are unchanged by this new analysis.

Reviewer #2 (Remarks to the Author):

Lundgren et al. reported an interesting study on substomatal cavity formation. When formation of stomatal complex was initiated but not completed in the mutant lines, substomatal cavity was not formed. This implies that the complete stomatal complex formation is prerequisite to the substomatal cavity formation. However, the authors did not follow the developmental processes of the stomatal complex or those of substomatal cavity. Because there is an age gradient along the grass leaf, the authors easily compare these developmental processes. Then, which is the first can be known.

In Arabidopsis thaliana plants, porosity of the palisade tissue was analysed, which is plausible. However, the stomatal conductance was measured for the whole leaf. Because the contribution of the stomata on the abaxial side of the leaf would be great, this reviewer wonders whether the authors are able to evaluate the stomatal conductance of the adaxial side. The stomatal conductance was measured in mature leaves. Thus, again, the sequence of the events is not clear.

This reviewer suggests that the authors should add some developmental data for wheat as well as for A. thaliana. This will clarify the sequence of the events.

We have now performed an analysis of the development of stomata and sub-tending airspace for both wheat and Arabidopsis (as requested by both this reviewer and reviewer 3). The data are shown in new main figure (Fig. 4) and supplementary figure (Fig. S4), using both CT and confocal microscopy, with a description of the new data in the text at lines 188-227 (wheat) and lines 253-261 (Arabidopsis).

To summarise these new findings, for wheat leaves the reviewer is absolutely correct that there is a developmental gradient from the base of the leaf (less differentiated) towards the tip (more differentiated). However, in the mature leaf 5 that we used for our experiments, even the stomata at the base of the leaf were fully differentiated so there was no gradient of stomatal density (Fig. S4a). However, in the transgenic TaEPF1-OE lines there was a decrease in stomatal density towards the base of the leaf which was paralleled by a decrease in mesophyll porosity (Fig S4b,c).

Although these data are consistent with our hypothesis of a link between stomata and mesophyll airspace formation, we decided to go further and to analyse much younger stages of leaf development (New Fig. 4). Unfortunately it is technically not possible to obtain gas exchange data at this resolution on such small leaves, but the microscopy data are consistent with our main hypothesis. Thus, the new results show that towards the tip of the wheat leaf there are mature stomata with large subtending airspaces, but that towards the leaf base there are immature stomata with no large subtending airspaces (Fig. 4a-f). Some very small airspaces are visible in the early mesophyll at cell junctions (marked with asterisks in Fig. 4e), so we cannot say absolutely that airspaces follow stomatal differentiation in wheat (they appear to form almost concurrently). Instead, the number and distribution of airspaces most easily fit to a hypothesis that small airspaces form at some mesophyll cell junctions as part of a normal developmental program, but with the subsequent size and extent of airspace formed being highly dependent on the formation of functional stomata in the overlying epidermis. In other words, proto-airspaces form, awaiting gaseous exchange through the stomata for full sub-stomatal development to occur.

Similarly in *Arabidopsis* (Fig. 4g-l), although small airspaces are visible at the mesophyll cell junction in the young leaves, these do not overtly correlate with the position of differentiating stomata, whereas in mature leaves there is an extensive network of airspaces, including large sub-stomatal cavities. It is technically not possible to obtain stomatal conductance data in these very small leaves at this resolution using standard gas exchange equipment.

Overall, the new developmental data indicate that airspaces form within the mesophyll in both wheat and *Arabidopsis* with a timing which is difficult to fully dissociate from stomatal differentiation. However, the pattern of mesophyll airspace initiation is not overtly linked to the pattern of stomatal initiation. The data are consistent with the idea that the subsequent degree and extent of mesophyll airspace formation is linked with the development of mature, functional stomata.

2) For *A. thaliana*, the adaxial stomatal data may be examined. The morphological estimate of stomatal conductance may be sufficient for this purpose.

Data on adaxial *Arabidopsis* stomata are now presented in Fig. 5i. These results are consistent with those previously published in a number of publications, i.e., *epf1/2* mutants display a very high stomatal density whereas overexpression of EPF leads to a decrease in stomatal density. With respect to g_{smax} , in the *focl-1* mutant (the key genotype in our experiment) it is very difficult to calculate this parameter since (as a consequence of the mutation) it is impossible to accurately measure the pore size (a key value in the calculation of g_{smax}). We believe that the measured value of g_s (shown in Fig. 5k) is much more meaningful since it clearly indicates the restricted gas exchange that occurs via the partially blocked stomatal pores observed in the *focl-1* mutant.

Minor points

L. 94: It is necessary to define width, area and length of stomatal complex. Perhaps, it is easy to use an illustration like the complex shown in Figure 1 d.

We now provide a diagram of a wheat stomatal complex in Fig S3d which defines the measurements taken in the analysis of stomatal size and g_{smax} .

L. 113: wider stomatal complex? greater stomatal complex??

This has been corrected

Figures: *P* may be in italic rather than in lower-case roman.

This has been addressed.

Figure S2: mesophyll surface area may be expressed on leaf surface area.

Although we can certainly express the exposed mesophyll surface area on a per leaf area basis, we think that expressing the data on a per volume basis makes more physiological sense: surely it is the surface area available for gas exchange over a volume of tissue that is most relevant to the exchange of material into and out of cells? We would prefer to remain with the figure (now Fig S3) as is, but can easily replace it with the data expressed per surface area if absolutely required by the reviewer/editor.

For Figure S2-a(b), which is the explanatory variables g_s or mesophyll surface (porosity)?

Our hypothesis is that g_s is the explanatory variable for mesophyll porosity in the figure (now Fig S3).

Reviewer #3 (Remarks to the Author):

This is an interesting paper that addresses the link between stomata and the underlying air-spaces in the mesophyll, a question that has important physiological and developmental relevance. The underlying concept is that because CO₂ must traverse the epidermis and the mesophyll air spaces in series, one might expect that their permeabilities are developmentally coordinated. The specific question addressed is whether it is the presence of stomata or their opening (and thus resulting gas exchange) that triggers air space development.

The study uses plant material with varied stomatal characteristics to show that there is a link between stomata and air-space development. The first figure uses different species and varieties of wheat (with varied ploidy) to demonstrate a correlation between g_s and mesophyll porosity. The transgenic OE of EPF1 in wheat was quite convincing: when stomata fail to differentiate, substomatal cavities do not form below them. However, (and as the authors' note) this could be due to a signal from the stomata or from their opening.

It is the Arabidopsis studies that the authors use to support the idea that it is gas exchange through stomata, and not just their presence, that is responsible for the linkage. Here the relation between stomatal conductance and air space porosity occurs only in the palisade mesophyll, and while the data are consistent with the authors' conclusions and indeed quite interesting, I wish that the paper did more to demonstrate a role for gas exchange in coordinating air space formation.

Specifically, the paper would be enhanced by clarification of the timing of the developmental and physiological events. With reference to % leaf expansion – when do stomata first become competent to open and when do the mesophyll cells separate? Could the authors do more to place these processes in a developmental context?

This point (which was also raised by reviewer 2) has been addressed with new data in Figs 4 and S4 and text on lines 188-227 (wheat) and lines 253-261 (Arabidopsis). A summary of the new data and its interpretation is provided in the response to Reviewer 2 (above).

Do we know that the four Arabidopsis lines compared in Figure 3 all followed the same patterns of leaf expansion and final size, thickness, etc.?

Descriptions of the growth characteristics all four Arabidopsis lines analysed in this paper have already been published by one of this paper's co-authors (Doheny-Adams et al. (2012) Philos. Trans, R, Soc. Lond B Biol Sci 367, 547-555: Hunt et al (2017) Plant Physiol. 174, 689-699). The lines were grown under the same conditions and in the same growth cabinets as the published data. The four lines grew in a very similar manner to that previously described, with no major differential patterns of growth observed that could serve as an explanation for the relationship of g_s and porosity. In our paper the Arabidopsis mutants are being used as well-characterised tools to test the hypothesis of a functional link between gas exchange (as measured by g_s) and mesophyll porosity (as measured by CT).

I also wondered why there were no experimental manipulations? It seemed to me that the system would be ripe for physiological experimentation and that this would help the paper arrive at a more convincing understanding of how such a "conductance-based mesophyll patterning system" might indeed operate.

We totally concur with the reviewer that experiments to directly manipulate the system, e.g., to physically alter stomatal conductance during the early stages of leaf development and observe the outcome on mesophyll porosity, would be really exciting. Indeed this is exactly what we attempted to do during the project. Unfortunately, we found these experiments to be technically extremely challenging. Gaining access to the leaf surface at the appropriate stage of development requires extensive dissection. Our lab has plenty of experience with this type of work, and accessing very young leaves of Arabidopsis was relatively straightforward. However, we found that any attempt to, e.g., coat the surface of the leaf at this stage to try and block the differentiating stomatal pores was always extremely deleterious to the further growth and development of the leaf, presumably due to loss of temperature control through evaporative cooling. This cessation of growth (or subsequent abnormal growth) happened very rapidly, so we think it would be very difficult (and inappropriate) to draw strong conclusions from any change in mesophyll differentiation observed in these samples. We will continue to try and devise novel ways to perform these types of experiments (they would be great to get working!) but we believe they are, unfortunately, simply technically not possible to do at present.

Reviewer #4 (Remarks to the Author):

This manuscript addressed an important developmental question in plant biology that has huge implications for plant physiological function, whether plants coordinate cuticular gas exchange with mesophyll porosity.

I liked the approach employed by the authors utilizing two species with very different leaf anatomies as well as mutants and the novel method of MicroCT to visualize the anatomy of the internal leaf.

Major comments:

- How can we rule out changes in cell size due to the ploidy changes driving the calculated porosity changes of the mesophyll in wheat? There seems to be a huge variation from the MicroCT scans of cell size that to my eye correlates with porosity. Cell size was not quantified and analyzed and should be, considering it is driven by genome size changes.

We now provide new data derived from confocal microscopy which gives measurements of mesophyll cell volume for 2n, 4n and 6n wheat lines (Fig S5f, lines 221-227). The data indicate that, as suggested by the reviewer, the mesophyll cells in the 6n lines are significantly larger than those in 4n and 2n lines. This fits with general observations on the

correlation of cell size and ploidy. However, one must be cautious about interpreting cell size data as a proxy for porosity since they are quite distinct measurements. Porosity is the inverse of the amount of solid tissue per organ volume. Conceptually, the solid tissue could consist of many small cells all tightly bound together or could consist of one giant cell, i.e., cell size can theoretically vary dramatically without any change in the porosity calculated. For example, in our previous work on Arabidopsis (Lehmeier et al, Plant J 2017) the transgenic line with one of the lowest leaf porosities contained mesophyll cells that were much smaller than normal. We believe that the change of cell volume between the different ploidy levels revealed by our new data is probably more significant for the outcome on surface area/volume relationships, which relates to other points raised by the reviewer (see below).

-Given the importance of mesophyll porosity for photosynthesis, please show the effect of these changes in porosity on photosynthesis. If these changes in mesophyll porosity are important there should be large differences in photosynthetic rate.

The reviewer raises an interesting point. The pattern of assimilation rate (A_{400}) across the different ploidy levels (Fig. S5a), the pattern of mesophyll conductance (g_m) across ploidy level (Fig. S5c), and the relationship of assimilation rate with porosity (Fig. S5e) are now shown in the new supplementary Figure S5. These data do not indicate any obvious relationship between porosity and photosynthesis across the different ploidy levels. Interestingly, however, when instantaneous water use efficiency (iWUE) is calculated, there is a significantly increased iWUE in the 6n lines compared to the 2n and 4n lines (Fig. S5b). These increases appear to mirror a decrease in exposed mesophyll surface area per volume as one proceeds from 2n through 4n to 6n lines (Fig. S5d). Finally, this decrease in tissue level surface area to volume ratio is the inverse of the increase in mesophyll cell volume measured as ploidy level increases (Fig. S5f). One interpretation of these data is that it was a drive towards increased water use efficiency, rather than photosynthetic assimilation, that has led to the changes in leaf structure reported in our paper. Specifically, if functional stomata are required to fully develop the sub-stomatal and mesophyll airspace, then these new data may point to the role of water vapour rather than CO_2 as the driver of this airspace development. This is, of course, speculation, but we believe it is a plausible idea, and one that will be of interest to people in the field. We thank the reviewer for raising this issue which has, we believe, led to increased potential impact of our work. The new data are described on lines 212-227.

-The authors constantly refer to porosity as an important driver of photosynthesis and gas exchange properties of the leaf yet never measured mesophyll conductance to CO_2 ? I imagine also that if mesophyll porosity as determined in this manner is important for photosynthesis then we ought to see a difference in mesophyll conductance. The authors have all of the equipment to measure this important trait.

This point is now covered in the previous section

-The TaEPF1-OE lines were an excellent system in which to show the presence of a fully developed stomata determines the formation of a substomatal cavity. Image d of Figure 3 is however at a very different plane to figure 2. Why the huge difference?

The paradermal plane in Fig 3d has been selected to be just below the stomata, thus addressing the issue of linkage to sub-stomatal airspace (which contributes to total mesophyll porosity). The paradermal images in Fig 2 are approximately half-way through the depth of the leaf and are, in essence, example images to give an idea of what the inside of the leaf looks like. The aim of Fig 2 is to provide quantitative data on the distribution of porosity across the whole depth of the leaves, as shown in Fig 2m,n,o (allowing comparison 2n, 4n, 6n)- a very different aim from Fig. 3. However, following this and other comments from the reviewer (below), we have amended Fig. 3 to include paradermal CT slices (Fig. 3f,g) which are more comparable (in terms of z-position in the leaf) with those shown in Fig. 2.

There is an obvious effect of a stomata on the presence of a substomatal cavity but how does that relate to mesophyll porosity as determined in Figure 2 jkl?

The most important data in Figure 2 are in parts m,n,o which show the porosity in every slice taken during CT analysis. These data provide quantitative data on porosity in each slice, some of which will incorporate sub-stomatal cavities. By integrating all the slices one can calculate the total porosity through the full depth of the leaf. The individual CT slices shown in Fig 2d-l serve to provide a visual impression of how the airspace/tissue is patterned in these individual slices. To repeat, these are exemplar 2D slices, whereas the values for porosity, exposed surface area per volume data, are all derived from the integrated 3D data of individual tissue samples.

In figure 2abc there is no layer clearly shown in these renditions that depicts the layer shown in Figure 3d. Yes I agree that stomata=substomatal cavity (a great result) but what about the mesophyll at the same height as calculated in Figure 2.

We have amended Fig. 3 to incorporate paradermal CT slices at a position similar to those shown in Fig. 2. The TaEPF1-OE leaves are visibly less porous at internal positions compared with controls (Fig 3f,g).

As indicated above, the 2D slices in Fig. 2d-l are to some extent arbitrary, enabling a quick visual comparison of the slices taken in different orientations (a major advantage of the CT approach since by classical sectioning of tissue researchers tend just to show cross-sections). It is possible to extract and display any CT slice, but obviously there is a size limit to a figure, plus showing massive number of slices in different orientations very quickly makes figures essentially unintelligible to most readers. All the CT data are stored and will be available as open access at the Hounsfield Facility in Nottingham post-publication, so individual readers will be able to access the data by this route, allowing them to analyse sub-domains of the CT leaf data as they require.

This is an inconsistency across this manuscript that porosity seems to be determined from a non-standard paradermal layer, either immediately below the epidermis (which would then just be a function of the number of stomata, an unsurprising result given sub-stomatal cavity is named for its known presence below stomata) and then in Figure 2 in the middle (??) of the leaf any correlation could then be due to simple changes in cell size but not leaf thickness. In addressing this concern please add images at the same plain of Figure 2 jkl for those of Figure 3 d and e.

As indicated above, all values of porosity given in this paper are integrations of data from many CT slices. For the purpose of providing visual data which are indicative of points we wish to raise (and to demonstrate the power of CT analysis when combined with physiological methods) we provide example images of individual CT slices taken in various orientations within the samples analysed for illustration. We must stress that the porosity values used in our analysis are never taken from individual CT slices. With respect to cell size, as indicated above, there is (conceptually) no direct, simple relationship of cell size and tissue porosity, therefore relating cell size to porosity needs to be done with caution. To address the point raised by the reviewer, we have now added images of CT slices from both WT and TaEPF1-OE lines to Fig. 3. These images (Fig. 3f,g) represent paradermal sections taken from approximately the middle of the leaf (in z), comparable to those shown in Fig. 2. Interestingly, the paradermal section from the TaEPF1-OE line suggests a denser tissue, consistent with the lower porosity values shown in Fig. 3i (which, to re-iterate, are integrated values from CT slices across the entire depth of different leaf samples).

Concerns with the method:

-Given that sections of transpiring leaves were taken for imaging under the MicroCT, how can we be certain that these differences in calculated mesophyll porosity are not due to changes in mesophyll cell size due to a drop in bulk cell turgor that happens when leaves are transpiring and under more negative water potentials? Just because the leaf section was enclosed in polystyrene doesn't mean that when the sample was excised the leaves didn't have different turgors due to different gs. I think that there might be very different results for non-transpiring, fully hydrated leaves, in this state we would get the true measure of comparative developmental differences in mesophyll anatomy. There could be an auto-correlation between mesophyll porosity in the higher transpiring leaves because they would have more negative water potentials, cell turgors and thus smaller mesophyll cells and greater porosity. There is lots of literature using cryosem showing that there are massive changes in mesophyll cell shape as leaf water potential declines, even slightly.

As indicated in Fig S5f, we now have measurements of cell volume for mesophyll cells for 2n, 4n and 6n wheat leaf samples. These new data indicate massive differences in mean cell volume (2n = c.2000 μm^3 , 4n = c.7000 μm^3 , 6n = c.10,000 μm^3), i.e., 6n cells are about 5 times larger than 2n cells. Of course, during the processing of samples for confocal analysis there will have been some change of volume, but it is difficult to believe that such massive changes in volume reflect a differential in volume change during processing for the different ploidy levels. All the 3D images we used indicated intact, turgid cells, with no evidence of collapse or "crinkling" (as is seen when cells undergo severe water loss during processing for microscopy). In addition, the CT analysis takes about 20 minutes per sample and any movement of the sample (even by 1 voxel) leads to image stacks which are blurry and essentially unusable. The fact that we obtain crisp, very high quality CT data indicates that cell shrinkage during analysis must be extremely minimal. Of course, it is very difficult to prove a negative and to totally discount any shift in cell/tissue volumes during processing, but we think that, realistically, these can only ever be very small in relation to the differences in porosity/tissue/cell volumes we are comparing in the different genetic backgrounds.

Minor comments:

Line 32: 'evolutionary significance' is a vague statement, considering in the same sentence the coordination of intercellular spaces and stomata is said to be essential for photosynthesis.

Amended

Line 109-115: to me this is an overreach - change like Line 143 to indirect. Genome size correlates with cell size so how can the authors claim that there has been active selection for cultivars with larger stomata. Isn't it more likely that domestication favoured larger cell sizes due to yield increases?

As indicated above, we favour an interpretation (based on the new data in Fig S5) that the observed altered stomatal size, mesophyll porosity, mesophyll exposed surface area and mesophyll cell size are more likely linked to altered water use efficiency than photosynthesis. We agree with the reviewer that any selection for these traits has been "indirect" and have amended the text accordingly.

Figures: Figure 2 Please mark with a dotted line on jkl where the transverse and longitudinal sections/slices were taken from and visa versa.

Figure and legend have been amended

Line 190: universality is a strong word - even with the caveat of potentially, considering only 2 angiosperm species were used. Note the work by Duckett and Presel in mosses and intercellular air spaces, this work should be consulted and referred to esp with regards to Lines 240-242.

We have now removed the term "universality" since we agree that we were probably over-reaching ourselves. As a consequence, we would prefer to keep the paper focussed on angiosperms. Expanding the paper to include data from evolutionary very distinct plants, although interesting for future work, risks over-complicating the story.

Figure 4. Does the focl-1 mutant have a substomatal cavity? Please check and discuss.

Yes it does. This is nicely shown in our previous paper (Hunt et al., Plant Physiol. 2016) and this point is now included in the text (line 277).

Figure 4I, again I would like to see how this difference in porosity relates to photosynthesis if that is why this is important.

As indicated in previous sections, we now include new data on wheat (Fig. S5) showing no obvious relationship between assimilation rate and porosity across the different ploidy levels. In contrast, as shown also in Fig S5, there appears to be a much stronger linkage with water use efficiency. With respect to Arabidopsis, our new analysis (Fig S7) suggests a similar situation in the eudicot leaf. Although there is a weak correlation of palisade porosity and assimilation rate (Fig S7a), the relationship between palisade porosity and water-use efficiency is much more striking (Fig S7b). Taking these data together (wheat of different ploidy, Arabidopsis mutants with different stomatal density/structure) we think the linkage of porosity to water-use (rather than photosynthesis) provides a novel insight into the system.

Reviewers' comments:

Reviewer #1 (Remarks to the Author):

All my previous concerns have been addressed satisfactorily. I have no further objections, nice study! Jaume Flexas

Reviewer #2 (Remarks to the Author):

The revised version with some additional data (particularly Fig. 4) reads well. This reviewer appreciates the efforts of the authors. This reviewer has only a few reservations.

When the relationship between the mesophyll surface area and the g_s in $\text{mol m}^{-2} \text{s}^{-1}$, the unit of the mesophyll surface area should be $\text{m}^2 \text{m}^{-2}$ (Figure S3). Perhaps, r^2 may not be significant. Then, the hypothesis in L. 163-164, "greater allocation of tissue volume to airspace and exposed surface area facilitates gaseous diffusion ...," should be modified. These features are more directly related to mesophyll conductance than stomatal conductance.

The trend that the mesophyll porosities in the 6n wheat lines small is clear. However, this cannot be directly related to low stomatal conductance.

L. 224-226: The hypothesis that during the selection of modern wheat there has been an increase in mesophyll cell size, with concomitant decrease in exposed surface area and decrease in mesophyll porosity, as well as altered stomatal parameters of increased size and decreased density, leading to improved water use efficiency. =>> This is misleading. As mentioned above, mesophyll surface area denotes the exposed mesophyll surface area per unit leaf area rather than the cell surface area / volume. The increase in mesophyll surface area per leaf area usually increases the A400, which would also contribute to the increase in iWUE.

Fig. S2 Paired g_s and porosity: This the scattered plot of the raw data shown in Fig. 1l. This reviewer cannot understand the title. May be paraphrased.

Fig. S5. The unit of A should be $\mu\text{mol m}^{-2} \text{s}^{-1}$. Mesophyll surface area per leaf area is better than the surface /volume.

Reviewer #3 (Remarks to the Author):

The authors have done a good job of addressing the issues raised by the reviewers and the resulting manuscript is much improved. While I would still have liked to have seen physiological experiments, the genetic approaches have value and support the authors' hypothesis. In particular, the EPFoe wheat plants are quite convincing (to me).

Reviewer #4 (Remarks to the Author):

I feel that the authors have done a good job at addressing my comments, it was not clear from the methods that the calculation of porosity was from the full stack of scans in 3D and not from just a simple single section. I am more comfortable now.

The units for A are wrong in both Figure S5a and FigureS7a, should be micromol

Lundgren et al Response to Referees

Reviewer 1, Reviewer 3 and Reviewer 4 are happy with the previous revised version of our manuscript, the only minor comment being the correction to the units for A (which should be μmol rather than mol) in Figure S5a and Figure S7a. This has been corrected.

The editorial team also spotted that in our *Source File* there was a mix up in the mesophyll surface area per volume data reported for Figs S3a and S5d. The numbers in the table were correct but had somehow been transposed to the wrong columns. This has been corrected in the revised *Source File* submitted with this version of the submission.

Reviewer 2 had a few points, which we have addressed, as described below.

The revised version with some additional data (particularly Fig. 4) reads well. This reviewer appreciates the efforts of the authors. This reviewer has only a few reservations.

i) When the relationship between the mesophyll surface area and the g_s in $\text{mol m}^{-2} \text{s}^{-1}$, the unit of the mesophyll surface area should be $\text{m}^2 \text{m}^{-2}$ (Figure S3). Perhaps, r^2 may not be significant. Then, the hypothesis in L. 163-164, "greater allocation of tissue volume to airspace and exposed surface area facilitates gaseous diffusion ...," should be modified. These features are more directly related to mesophyll conductance than stomatal conductance.

In the **revised Supplementary Figure S3b** we now provide an analysis of mesophyll surface area per leaf surface area vs g_s , as requested by the reviewer. These data indicate that there is a relationship between these two variables. The strength of the relationship ($r^2 = 0.633$) is not quite as strong for that observed when mesophyll surface area per volume is plotted against g_s ($r^2 = 0.718$, Fig. S3a) and the confidence that can be assigned is slightly lower ($P = 0.0323$ vs $P = 0.0161$), but nevertheless we think that our conclusion on line 163 that "greater allocation of tissue volume to airspace and exposed surface area facilitates gaseous diffusion ...," remains a reasonable interpretation of the data. To summarise, we now include a new data analysis in Fig S3 (reported in the text at L155-157) which addresses the point raised by the reviewer. We have slightly amended the text L163-5 to make the point clearer.

ii) The trend that the mesophyll porosities in the 6n wheat lines small is clear. However, this cannot be directly related to low stomatal conductance.

We agree with the reviewer that when it gets to the resolution of comparing different cultivars/lines within a ploidy level our data are not sufficiently extensive to make any strong conclusions (essentially $n = 2$ or 3) and we have tried to avoid suggesting such a "within-ploidy" relationship in the manuscript. Clearly stomatal conductance is influenced by many factors, not simply ploidy, and deciphering these influences is an important area for ongoing and future research. Our view is more that major changes in ploidy during seem to have shifted the baseline against which these other factors act to influence stomatal conductance.

iii) L. 224-226: The hypothesis that during the selection of modern wheat there has been an increase in mesophyll cell size, with concomitant decrease in exposed surface area and decrease in mesophyll porosity, as well as altered stomatal parameters of increased size and decreased density, leading to improved water use efficiency. =>> This is misleading. As mentioned above, mesophyll surface area denotes the exposed mesophyll surface area per unit leaf area rather than the cell surface area / volume. The increase in mesophyll surface area per leaf area usually increases the A400, which would also contribute to the increase in iWUE.

How cell size and shape has altered during wheat leaf evolution, and how this relates to bulk function of the leaf in terms of gas exchange and photosynthesis is a fascinating issue which our research is currently addressing. The topic and conclusions go beyond what we think can reasonably be addressed in this manuscript, but below we give a summary of what our data indicate.

Firstly, although the reviewer correctly suggests that there is often a relationship between A400 and exposed mesophyll area per leaf area, this is not a universal relationship. Indeed, if we analyse A400 and the mesophyll area per leaf area using our data presented in Fig. S3 on mesophyll exposed surface area per leaf area and A400 (Fig S5a) there is no evidence of a correlation ($r^2 = 0.104$, $P = 0.48$). This is in stark contrast to the relationship shown in Fig S3b between mesophyll exposed surface area per leaf area and g_s ($r^2 = 0.633$, $P = 0.0323$). At the cell level, we believe that with increasing ploidy the leaf faces a conundrum. As ploidy increases, cell volume generally increases (this has long been observed, although the mechanistic cause remains debated). As cell volume increases, then cell surface area/volume will decrease- if cell shape and cell separation remain the same. If a cell needs to maintain a certain gas flux, then there needs to be a change of cell shape and/or cell-cell separation, which will have knock-on effect on, e.g., cell density, air space and associated parameters of gas flux, which the plant needs to balance out at the whole leaf level. In summary, we see this as a very complicated evolutionary geometric challenge for the leaf in which cell size and shape parameters need to be integrated with physiological requirements at both a local cell and global whole organ level. The present data in the field is, to our mind, generally insufficient to properly address this issue but we think that the combination reported here of cell/tissue imaging with gas exchange analysis opens the door to future work to tease out these relationships. Our suggestion on lines 224-226 should perhaps be seen more as a plausible speculation to stimulate the field rather than a statement of fact that this is how the system works. We have amended the text in Lines 223-229 to make this point clearer.

Fig. S2 Paired g_s and porosity: This the scattered plot of the raw data shown in Fig. 1l. This reviewer cannot understand the title. May be paraphrased.

This has been altered on the figure and in the summary of supplemental material (line 442-3).

Fig. S5. The unit of A should be $\mu\text{mol m}^{-2} \text{s}^{-1}$. Mesophyll surface area per leaf area is better than the surface /volume.

This has been corrected.